# Genomic Structural Diversity in Local Goats: Analysis of Copy-Number Variations

**DOI:** 10.3390/ani10061040

**Published:** 2020-06-16

**Authors:** Rosalia Di Gerlando, Salvatore Mastrangelo, Angelo Moscarelli, Marco Tolone, Anna Maria Sutera, Baldassare Portolano, Maria Teresa Sardina

**Affiliations:** 1Dipartimento Scienze Agrarie, Alimentari e Forestali, University of Palermo, 90128 Palermo, Italy; rosalia.digerlando@unipa.it (R.D.G.); salvatore.mastrangelo@unipa.it (S.M.); angelo.moscarelli@unipa.it (A.M.); marco.tolone@unipa.it (M.T.); baldassare.portolano@unipa.it (B.P.); 2Dipartimento Scienze Veterinarie, University of Messina, 98168 Messina, Italy; asutera@unime.it

**Keywords:** copy number variation, genotyping array, Sicilian goat breeds

## Abstract

**Simple Summary:**

Copy-number variations (CNVs) are one of the widely dispersed forms of structural variations in mammalian genomes and are known to be present in genomic regions that regulate important physiological functions. In this study, CNV detection was performed starting from genotypic data of 120 individuals, belonging to four Sicilian dairy goat breeds, genotyped with the Illumina GoatSNP50 BeadChip array. Using PennCNV software, a total of 702 CNVs were identified in 107 individuals. These were merged in 75 CNV regions (CNVRs), i.e., regions containing CNVs overlapped by at least 1 base pair. Functional annotation of the CNVRs allowed the identification of 139 genes/loci within the most frequent CNVRs, which are involved in local adaptation, mild behaviour, immune response, reproduction, and olfactory receptors. This study provides insights into the genomic variations within these Italian goat breeds and should be of value for future studies to identify the relationships between this type of genetic variation and phenotypic traits.

**Abstract:**

Copy-number variations (CNVs) are one of the widely dispersed forms of structural variations in mammalian genomes, and are present as deletions, insertions, or duplications. Only few studies have been conducted in goats on CNVs derived from SNP array data, and many local breeds still remain uncharacterized, e.g., the Sicilian goat dairy breeds. In this study, CNV detection was performed, starting from the genotypic data of 120 individuals, belonging to four local breeds (Argentata dell’Etna, Derivata di Siria, Girgentana, and Messinese), genotyped with the Illumina GoatSNP50 BeadChip array. Overall, 702 CNVs were identified in 107 individuals using PennCNV software based on the hidden Markov model algorithm. These were merged in 75 CNV regions (CNVRs), i.e., regions containing CNVs overlapped by at least 1 base pair, while 85 CNVs remained unique. The part of the genome covered by CNV events was 35.21 Mb (1.2% of the goat genome length). Functional annotation of the CNVRs allowed the identification of 139 genes/loci within the most frequent CNVRs that are involved in local adaptations, such as coat colour (*ADAMTS20* and *EDNRA*), mild behaviour (*NR3C2*), immune response (*EXOC3L4* and *TNFAIP2*), reproduction (*GBP1* and *GBP6*), and olfactory receptors (*OR7E24*). This study provides insights into the genomic variations for these Sicilian dairy goat breeds and should be of value for future studies to identify the relationships between this type of genetic variation and phenotypic traits.

## 1. Introduction

Copy-number variations (CNVs) are structural variations widely dispersed in mammalian genomes. Conventionally, CNVs are defined as genomic regions ranging from 50 base pairs (bp) to several Mega bases (Mb), and which are present in the form of deletions, insertions, or duplications [1,2]. CNVs have been shown to contribute to phenotypic diversity in model organisms and are thus associated with genetic diversity as well as important production and disease traits in domesticated livestock species [3,4,5]. CNVs involving large genomic regions can affect the gene structure and gene dosage, which, in turn, has an impact on the gene expression, influencing in this way phenotypic variability and disease susceptibility [1,6,7,8]. 

Several techniques have been developed and used to identify CNVs within the genome, including comparative genomic hybridisation arrays, SNP genotyping arrays, and next-generation sequencing [9,10]. SNP arrays provide the Log R Ratio and B allele frequency, which represent the intensity signal due to the copy number and allelic status of the population [11]. Different algorithms have been used for identification of CNVs from molecular data [12]. For example, the CNV discovery method of the PennCNV software tends to focus on finding long and/or rare CNVs while other methods, like the multivariate one, considering all samples simultaneously, was designed for detecting small and common CNVs [12,13]. However, PennCNV is the most reliable and accurate algorithm in detecting CNVs from Illumina BeadChip data [14,15,16,17]. It is also worth pointing out that although the SNP arrays can be used to detect the CNVs, the SNP probes on the chip are neither dense enough nor uniformly distributed to achieve an unbiased high resolution CNV map [18,19].

Previous studies have completed genome-wide CNV analysis in many livestock species, including cattle [19,20,21,22,23,24], sheep [25,26,27,28,29,30], pigs [31], and horses [32]. However, in goats, only few studies have been conducted based on SNP array data (e.g., [13,33]).

Since CNVs are known to be present in genomic regions that regulate important physiological functions, they have also been studied for associations with environmental adaptation and economically important traits in livestock species. In sheep, goats, pigs, and cattle, coat colour is partially determined by CNVs [34,35,36,37]. In cattle, traits and diseases, such as milk production [5,38,39] and female fertility failure [40], have been shown to be influenced by CNVs. Zhang et al. [41] suggested that CNVs may contribute to different litter size traits in Laoshan dairy goats. Di Gerlando et al. [30] found CNVs associated with milk production and health-related traits in dairy sheep. Qanbari et al. [42] reported duplication of the olfactory receptor family genes within the bovine genome, suggesting their role in environmental adaptability.

During the last century, erosion of livestock genetic resources was observed as a result of the massive replacement of low-productivity local breeds with highly productive ones [43]. In Sicily, dairy goat production represents an important resource for the economy of the hilly and mountainous areas, in which other economic activities are limited. Nowadays, four native dairy goat breeds are reared: Argentata dell’Etna, Derivata di Siria, Girgentana, and Messinese. These breeds present differences in both phenotypic and production traits. Selection breeding schemes are almost absent within Sicilian goat farms. Nevertheless, these breeds are adapted to the harshness of the mountain areas because of their good grazing characteristics, have an aptitude for dairy production, and are resistant to environmental conditions. Their adaptability is also expressed in longevity, resistance to diseases, and good fertility [43]. As reported above, different studies have been conducted on CNV detection in livestock species, but several local breeds still remain uncharacterized since their CNV distribution in the genome has not yet been analysed; for example, the Sicilian goats. Therefore, this study aimed to investigate the CNVs in these four dairy goat breeds by performing a genome-wide screen of autosomes using genomic data generated from the GoatSNP50 BeadChip Genotyping array (Illumina). The genome-wide analysis and characterization of the CNVs could improve our understanding on genetic variations, and could be an important tool for determining the role of CNVs on economic traits and for the development of conservation programs for these local breeds. 

## 2. Materials and Methods

### 2.1. Samples and Genotyping

A total of 120 individual blood samples were collected from four Sicilian goat breeds: Argentata dell’Etna (*n* = 24), Derivata di Siria (*n* = 36), Girgentana (*n* = 36), and Messinese (*n* = 24).

The Girgentana goat in one of the ancient Sicilian autochthonous breed, with long corkscrew horns in both sexes and a cream/white coat colour; the Derivata di Siria breed (also known as Mediterranean Red) is completely red with long ears. The Messinese and Argentata dell’Etna are reared in the mountain areas; the former presents a very different coat colour, e.g., plain, pied or streaked, black, brown, or red with various shades; the latter has a coat colour that has grey shading from light to dark, with silver glints, and a grey skin. All of them are dairy goat breeds and their milk is used for niche products in local markets (http://eng.agraria.org/).

All the procedures were approved by “Organismo Preposto al Benessere Animale” (O.P.B.A) of the University of Palermo in agreement with the recommendations of the European Union (EU) Directive 2010/63/EU to ensure appropriate animal care. About 6 mL of blood were collected from the jugular vein using tubes with EDTA as an anticoagulant. Genomic DNA was extracted from the buffy coats of the nucleated cells using the salting-out method [44].

Samples were genotyped using the Illumina GoatSNP50 BeadChip Genotyping array containing 53,347 SNPs. SNPs were mapped using the goat assembly ARS1 (RefSeq assembly accession: GCF_001704415.1, ftp://ftp.ncbi.nlm.nih.gov/genomes/all/GCF/001/704/415/GCF_001704415.1_ARS1). GenomeStudio v2.0 software (Illumina) was used to create the genotypic and intensity data for further analyses. 

### 2.2. Genetic Relationship among Sicilian Goat Breeds

Genotypic data was edited using PLINK 1.7 [45]. Only SNPs located on autosomes were considered. SNPs were filtered according to quality criteria that included a call frequency (proportion of samples with the genotype at each locus) ≥ 0.98, minor allele frequency (MAF) ≥ 0.05, and Hardy–Weinberg equilibrium (HWE) *p* < 0.001. SNPs and samples that did not satisfy these quality criteria were excluded. Pair-wise genetic relationships were estimated using the --cluster and --mds-plot options in PLINK 1.7 [45], and graphically represented by multidimensional scaling (MDS) analysis.

### 2.3. Identification of CNVs and CNVRs

The PennCNV software [46] was used to infer CNVs in our samples. The input file contained the signal intensity ratios (Log R Ratio, LRR) and allelic frequencies (B Allele Frequency, BAF) for 50,619 autosomal SNPs and 120 samples. The CNVs were inferred using the hidden Markov model [46], spanning at least three SNPs, using the population frequency of the B allele (PBF) file calculated on the BAF of each SNP, and considering the GC content within 1 Mb surrounding each SNP for the appropriate LRR adjustments. PennCNV quality filters were applied in order to keep high-quality samples: standard deviation of LRR < 0.3, BAF drift < 0.01, and waviness factor value between −0.05 and 0.05. CNV regions (CNVRs) were defined as suggested by Redon et al. [47], i.e., regions containing CNVs overlapped by at least 1 bp, and were identified using the *merge* command of BEDTools software [48]. Finally, CNVRs were conventionally defined as gain, loss, and mixed, i.e., CNVRs containing duplication, deletion, or both duplication and deletion CNV events. 

Additionally, a Venn diagram was constructed by an online tool (http://bioinformatics.psb.ugent.be/webtools/Venn/) using the CNVRs identified among the four considered breeds. Moreover, in order to highlight the differences among breeds related to the CNVRs, a matrix was also built assigning “0″ to the absence of a CNV in a CNVR (normal state), “1″ to a deletion, or “2″ to a duplication event. This matrix was used to perform a Principal Components Analyses (PCA) using PAST 3.22 software [49].

### 2.4. Gene Enrichment and Functional Annotation

The gene content was assessed for CNVRs with a frequency > 0.10 (i.e., present in at least 11 samples). It was described based on the *Capra hircus* genome assembly ARS1, in the Genome Data Viewer genome browser (https://www.ncbi.nlm.nih.gov/genome/gdv/browser/?context=genome&acc=GCF_001704415.1). The gene list was further analysed with the PANTHER Classification System (http://www.pantherdb.org/) to identify the gene ontology (GO) terms for the molecular function, biological process, cellular component, and biological pathway. It is worth noting that a portion of the genes in the goat genome has not been annotated or has been annotated as LOC, and this could influence the outcome of the GO analysis. For this reason, the GO enrichment analysis was conducted considering bovine genes.

## 3. Results and Discussion

In the present study, we described the CNVs in four goat breeds: Argentata dell’Etna, Derivata di Siria, Girgentana, and Messinese. These are local dairy breeds that are often replaced by highly productive foreign breeds, leading to the progressive abandoning of low-income rural activities [43].

### 3.1. Genetic Relationship among Sicilian Goat Breeds

A total of 49,234 SNPs and 120 individuals were used in this analysis. Multidimensional scaling analysis was performed to cluster the animals and to explore the genetic relationship among the Sicilian breeds. The obtained results evidenced that the Girgentana (Gir) and Derivata di Siria (DdS) breeds are distinct groups (Figure 1), as reported in previous studies [43,50,51], while the Argentata dell’Etna (Arg) and Messinese (Mes) individuals are positioned close to each other (Figure 1). The genetic closeness between Arg and Mes could be explained considering their geographic proximity, similarities in environment, breeding practices, and the gene flow among them.

### 3.2. CNV and CNVR Identification among Breeds

Using PennCNV, we were able to identify a total of 702 CNVs (Appendix A) in 107 individuals. Among the detected CNVs, 500 were gains and 202 losses, with an average number of 6.5 CNVs per individual and an average length of 184.2 kb per CNV. Similar results (i.e., 6.2 CNVs per individual) were reported by Liu et al. [33], who studied the CNVs in worldwide goat populations, also using the Illumina GoatSNP50 BeadChip array. The highest number of CNVs (*n* = 70) was found on chromosome 10, while only one CNV was identified on chromosomes 2, 20, 24, and 25. No CNV was found on chromosome 22. Merging the overlapping CNVs, 85 CNVs remained unique, i.e., present in only one individual (Appendix A), while a total of 75 CNVRs were identified among the breeds (Appendix A). The part of the genome covered by CNV events (CNVRs and unique CNVs, together) was 35.21 Mb, which corresponded to 1.2% of the *Capra hircus* genome length. The length of the CNVRs varied from 49.4 kb (CNVR_21) to 1.98 Mb (CNVR_16) (Appendix A), with an average length of 282.94 kb. All these statistics are consistent with the study of Liu et al. [33]. Among the 75 CNVRs, 42 were gains, 22 were losses, and 11 were mixed (i.e., containing duplication, deletion, or both duplication and deletion CNV events) (Appendix A, Figure 2).

Table 1 includes the CNVR details for each breed. It is possible to note that the Arg breed had the highest number of CNVRs (*n* = 48) and the highest number of gain events (*n* = 40) compared to the other breeds. Mes was the breed with the lowest number of CNVRs (*n* = 41), and DdS was the breed with the lowest number of gain events (*n =* 24).

The Venn diagram reported in Figure 3A show the number of CNVRs shared among the four breeds (*n* = 14) (Appendix A). Moreover, the PCA analysis performed using the matrix of the CNVRs (Appendix A; Figure 3B) show two separated groups for Gir and DdS, and closer ones for Arg and Mes. The highest number of shared CNVRs is between Gir and DdS (*n =* 10), followed by the ones shared between Arg and Mes (*n =* 8), which could be the reason for the partial overlap among these breeds presented in the PCA graph.

### 3.3. CNVR Gene Contents and Functional Annotations

Gene enrichment was described for the most representative CNVRs (frequency > 0.10; i.e., present in at least 11 samples) identified among the breeds. A total of 13 CNVRs presented frequency values between 0.103 and 0.430, and contained 139 genes/loci, completely or partially overlapping the CNVRs (Appendix A). To further analyse the biological functions of the CNVRs, we conducted a GO analysis. Similar to the studies on CNVs carried out on other livestock species (e.g., [19,23,24,28,30,32]), the results showed that the functions of the proteins encoded by these genes included several biological processes, molecular functions, cellular components, and pathways, as reported in Table 2. Among the most representative CNVRs, many of them included candidate genes associated with important breeding traits.

The CNVR_11 and CNVR_54 contained genes previously described as related to coat colour traits and showing CNV in goats [33,52]. The CNVR_11 partially contained the *ADAMTS20* gene and presented the highest frequency value (0.430). The *ADAMTS20* (ADAM Metallopeptidase with Thrombospondin Type 1 Motif 20) gene is involved in binding (GO:0005488), catalytic activity (GO:0003824), and molecular function regulator (GO:0060089); in the cellular process (GO:0009987) and in extracellular region (GO:0005576) (Table 2). It was reported that this gene had a role in coat colour variation during goat domestication [53] and it was associated with melanocyte development [54]. Moreover, Liu et al. [33] reported for the first time the *ADAMTS20* CNV in goats, with a high frequency (0.4974) within the Eastern Mediterranean (EME) goat breeds. The presence of this gene within the CNVR detected in our breeds confirmed their results, considering that Arg, DdS, and Gir belong to the EME group. 

The CNVR_54 (frequency value 0.280) contain the *EDNRA*, *TMEM184C*, *PRMT9*, *ARHGAP10*, and *NR3C2* genes, which were mapped on a 1 Mb CNV in Boer goats [52]. Among them, the *EDNRA* and *ARHGAP10* genes were present in specific pathways (P00019, P00034, and P00047), and together with *NR3C2* were involved in biological regulation (GO:0065007), metabolic process (GO:008152), and several molecular functions (GO:0005488, GO:0003824, GO:0098772, and GO:0140110). The duplication of the *EDNRA* (endothelin receptor type A) gene was positively associated with the degree of white spotting in goats [52], while the variation in the number of copies of the *NR3C2* (Nuclear Receptor Subfamily 3 Group C Member 2) gene was related to adaptation traits to harsh and dry environments; moreover, this gene is also involved in the very mild behaviour of the Gir goat breed [55].

Interestingly, among the 30 individuals carrying this CNVR_54 (Appendix A), the ones belonging to the solid-coloured DdS breed were absent, as well as 28 animals belonging to the Gir breed. 

The CNVR_33, CNVR_65, and CNVR_7 (frequency values of 0.411, 0.336, and 0.280, respectively) overlapped, with genes related to the immune system and belonging to several GO terms (Table 2). In particular, CNVR_33 contain the T-cell receptors alpha chain V, which are important components in the adaptive immune system and that were described as the CNVs in Criollo Argentino horses [32] and in Brown Swiss dairy cattle [5]. The CNVR_65 contain the *EXOC3L4* (exocyst complex component 3 like 4) and *TNFAIP2* (TNF alpha induced protein 2) genes, which are associated with natural antibodies and considered as candidate genes for innate host defence and disease resistance in dairy cows [56]. These genes are involved in the binding function (GO:0005488) and in cellular and localization processes (GO:0009987, GO:0051179). The CNVR_7 contained several guanylate-binding proteins (*GBP-1, GBP-4, GBP-5, GBP-6*) that are proteins involved in the inflammatory process and host defence against cellular pathogens in mammals (GO:0002376 and GO:0050896) [57]. Moreover, the CNVR_25 presented the *OR7E24* (olfactory receptor 7E24-like) and putative gustatory receptor genes, which are important to detect and identify a wide range of odours and chemosensory stimuli. It was reported by some authors [19,58,59] that these genes are duplicated within the bovine genome, suggesting their role in local adaptation. All these genes related with the immune system, host defence against pathogens, and olfactory receptors are consistent with the phenotypic characteristics of these breeds, which are well adapted to harsh environments and are resistant to infectious diseases. 

Finally, it is interesting to note that the *GBP1*, *GBP6* (on CNVR_7), and *SLC7A3* (cationic amino acid transporter 3) (on CNVR_59) genes found by Zhang et al. [51] as CNVs were associated with high fertility and litter size traits in goats. These are noteworthy traits in Sicilian goat breeding farms, where goats often gave twin births and kids are used for meat production. 

Although studies on goat CNVs have been previously published [33,43,51], this is the first study on the detection of copy number variations using the Illumina GoatSNP50 BeadChip array in Sicilian local goat breeds. It could be difficult to directly compare the CNV results across different studies. The use of different algorithms, different technologies, different criteria for filtering data, and differences in the numbers of samples and breeds tested could be responsible for the discrepancies. Moreover, all these factors could lead to a lack of overlap among the different studies on the same species. Nevertheless, our results could be considered reliable since the representation of the CNVRs in coat colour traits, immune related genes, and olfactory receptors was reported previously in livestock species [5,19,32,33,43,46]. Moreover, a biological link to traits, such as reproduction and behaviour, the immune response, and resistance/susceptibility to infectious diseases, which are known to be under selection, can be inferred for most genes within the CNVRs, and as reported above, the goat breeds involved in this study are well adapted to harsh environments and are resistant to infectious diseases. 

## 4. Conclusions

Additional analysis based on sequencing would help confirm our findings to further enhance the power to identify additional CNVs and genes in these goat breeds. Moreover, further studies would allow determining whether the CNVs could influence phenotypic variation, adaptiveness, and susceptibility/resistance to diseases in cases where their presence alter gene dosage, and may serve to improve the conservation and breeding programs for these local breeds.

## Figures and Tables

**Figure 1 animals-10-01040-f001:**
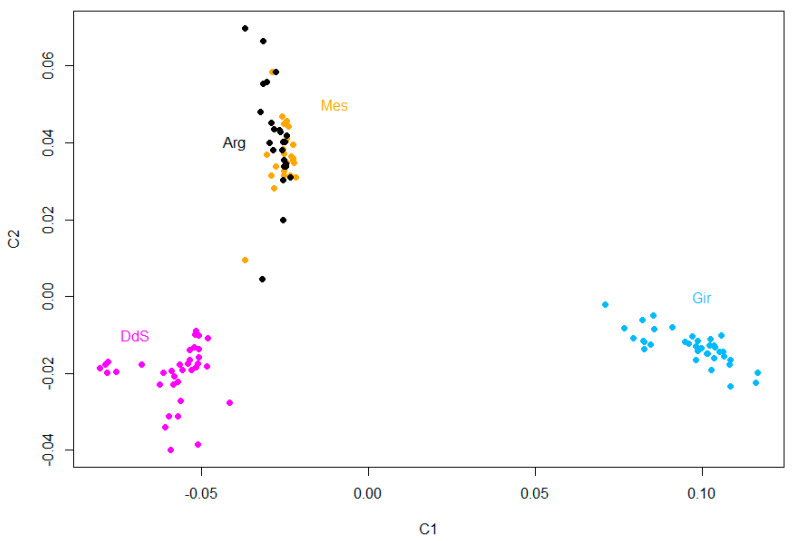
MDS plot of the Sicilian goat breeds. Arg = Argentata; DdS = Derivata di Siria; Gir = Girgentana; Mes = Messinese.

**Figure 2 animals-10-01040-f002:**
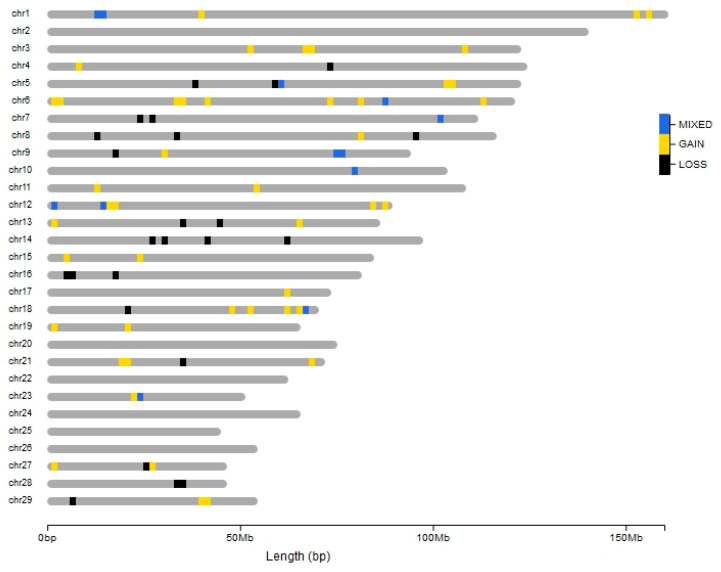
Ideogram of the distribution of the copy-number variation regions (CNVRs) in Sicilian goat breeds. Blue, yellow, and black colours represent mixed, gain, and loss events, respectively.

**Figure 3 animals-10-01040-f003:**
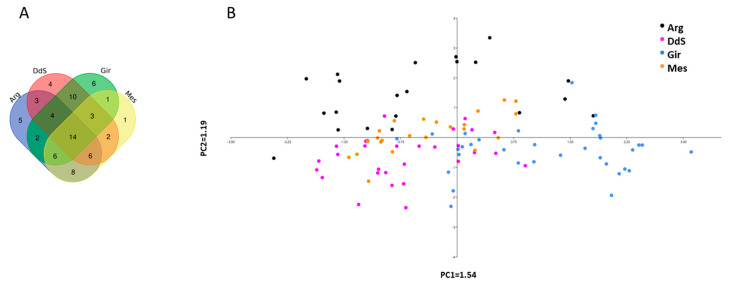
(**A**) Venn diagram of shared and proper CNVRs of each breed. (**B**) Principal Component Analysis obtained using the CNVR matrix 0, 1, 2 (0 = normal state; 1 = deletion; 2 = duplication). Arg = Argentata; DdS = Derivata di Siria; Gir = Girgentana; Mes = Messinese.

**Table 1 animals-10-01040-t001:** CNVR details for each goat breed and overall. Number of animals per breed are reported within brackets. Arg = Argentata; DdS = Derivata di Siria; Gir = Girgentana; Mes = Messinese.

Breed	No CNVRs	Gain	Loss	Mixed
Arg (*n* = 21)	48	40	6	2
DdS (*n* = 29)	46	24	18	4
Gir (*n* = 34)	46	27	17	2
Mes (*n* = 23)	41	30	8	3
Overall (*n* = 107)	75	42	22	11

**Table 2 animals-10-01040-t002:** The gene ontology (GO) categories (biological process, molecular function, and cellular component) and pathways containing genes overlapping the most frequent CNVRs identified in Sicilian goat breeds.

Accession Number	Biological Process	Gene Symbol
GO: 0065007	Biological regulation	EDNRA, NR3C2
GO: 0009987	Cellular process	ADAMTS20, EXOC3L4, ERG, KYAT3, LGALS9, TNFAIP2
GO: 0002376	Immune system process	GBP4, GBP5, GBP6
GO: 00051179	Localization	EXOC3L4, KCNJ15, SLC7A3, TNFAIP2
GO: 0008152	Metabolic process	ERG, KYAT3, NR3C2
GO: 0032501	Multicellular organismal process	OR7E24
GO: 0050896	Response to stimulus	GBP4, GBP5, GBP6
**Accession Number**	**Molecular Function**	**Gene Symbol**
GO: 0005488	Binding	ADAMTS20, EXOC3L4, GBP4, GBP5, GBP6, LGALS9, NR3C2, TNFAIP2
GO: 0003824	Catalytic activity	ADAMTS20, ARHGAP10, GBP4, GBP5, GBP6, KYAT3
GO: 0098772	Molecular function regulator	ADAMTS20, ARHGAP10
GO: 0060089	Molecular transducer activity	OR7E24
GO: 0140110	Transcription regulator activity	ERG, NR3C2
GO: 0005215	Transporter activity	KCNJ15, SLC7A3
**Accession Number**	**Cellular Component**	**Gene Symbol**
GO: 0005623	Cell	EXOC3L4, KYAT3, TNFAIP2
GO: 0005576	Extracellular region	ADAMTS20
GO: 0043226	Organelle	ERG
**Accession Number**	**Pathway**	**Gene Symbol**
P00019	Endothelin signalling pathway	EDNRA
P00034	Integrin signalling pathway	ARHGAP10
P00047	PDGF signalling pathway	ARHGAP10, ERG

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
