# Peer review of "Genomic Structural Diversity in Local Goats: Analysis of Copy-Number Variations"

_animals, 2020, doi:10.3390/ani10061040_

Round 1
Reviewer 1 Report
The manuscript entitled “Genomic structural diversity in local goats: Analysis of copy number variations” use a known approach to provide insights on CNVs in four Sicilian goat breeds. Subject is interesting but manuscript as its present form is not acceptable. Language needs to be checked and corrected.
The introduction is very difficult to read as it currently stands as sentences are long, poorly structured and without clear meaning. I suggest the entire manuscript is revised by a fluent English speaker before resubmission.
I suggest the authors include in the introduction a comment on the limitations of detecting CNVs in goats, and in local breeds.
In addition, I suggest the authors perform principal component analysis between all samples so the readers can visibly see the genetic relationships among the breeds.
In principle, analyses are made correctly but manuscript is not clear, methods are explained poorly and discussion part needs to be reformulated.
Some detailed comments are presented below but the most crucial thing is to make manuscript more consistent and clear.
Simple summary and abstract
1) Authors mention the molecular tool to detect CNVs and results, but they skipped mentioning the approach they´ve used to detect those CVNs.
I also miss in this summary to get some information about if authors pursue some goals on detecting differences among those local breeds, and what are those breeds selected for, meat, milk, double-purpose…
- Line 16 ” a total of 702 CNVs were identified in 108 individuals. These were merged in 75 CNV regions.”
I suggest to improve redaction of these two paragraphs, indicating the threshold set to define those regions.
3) Functional annotation of the 139 genes clustered only on those functions? Or is it that authors selected functions derived of what may be related with their breeds? It would give some clarity for readers if authors can be more precise on their methodology.
4) Line 35: “ this study provide insights of genomic variations within these Italian goat breeds……”
Introduction
Line 41-43 : “Copy number variations are the results of rearrangements of the genome”
I suggest to improve this first paragraph. It is too short, it gives no precise information and lacks references. Instead, authors can explain what type of rearrangements. And please define acronyms before mentioning them, i.e “base pairs (bp) and mega base (Mb)
Lines 44-48: This phrase is confusing. First, authors say that SNPs are widely used on genetic variation studies and then their statement is, that because of the coverage within the genome, CNVs are promising to detect stronger effects on gene regulation and expression.
To my opinion, one thing are genetic variation analysis and other is an analysis to detect gene regulation and expression. Then, the size of the CNVs is not necessarily correlated to its “stronger effect”, maybe what they meant was their significance on detection of variations on those “regulation and expression” regions, although, this is not the case either.
Line 53. “However, in goat, only few studies have been conducted based on SNP array data”
I suggest a deeper literature searching on CNVs, for example:
- An initial comparative map of copy number variations in the goat (Capra hircus) genome
- Copy-number variation in goat genome sequence: A comparative analysis of the different litter size trait groups.
- Genome-wide CNV analysis revealed variants associated with growth traits in African indigenous goats.
Lines 63-67 please provide references.
Material and methods
Samples and genotyping. Please provide information on the DNA extraction methodology, also specify the provider of the goat genome assembly (ensemble, ucsc….). I also missed information of the filtering methodology to obtain the final dataset and the software(s) used.
Identification of CNVs and CNVRs Line 84-89 this paragraph is too long and complex t oread, please rephrase.
I also do not fully understand authors CNVs catalogue “gain, loss and complex”, based on the previous criteria? Whatsoever… this section needs to improve its redaction.
Results and discussion
Lines 110-112: please provide references
Lines 149-150 “ The presence of this gene 149 within the CNVR detected in our breeds confirm their results, considering that Girgentana, Derivata di Siria …….belong to the EME group”
Table 1 Please give some format to the table, there are no legends on “biological function” or the category under that column or “gene names”
Lines 167-178 are more results. I miss some discussion on these results based on what authors know about these breeds
Author Response
Reviewer 1

Reviewer 2 Report
The manuscript identified 75 CNVRs and 139 genes/loci related to phenotypic traits by the PennCNV software and further functional enrichment analysis using the goatSNP50 BeadChip data of 120 samples belonging to four local goat breeds. This study supplied the knowledges of the relationship between genomic variations and phenotypic traits.
However, the workload cannot meet the publication requirement. The method used in this manuscript is just an elementary research on CNV analysis. Furthermore, the CNVs identified by chip data always present high false positive for its low resolution. From the perspective of innovation, there was no new approach and significant findings in this study.
The results can be further improved by adding the comparison of the breed-differential CNVs to reveal the breed differences and qPCR validation of CNVs. The association analysis between the CNVs and phenotypic traits can also be included in this manuscript if the phenotype data available.
Author Response
Reviewer 2
Comments and Suggestions for Authors
The manuscript identified 75 CNVRs and 139 genes/loci related to phenotypic traits by the PennCNV software and further functional enrichment analysis using the goatSNP50 BeadChip data of 120 samples belonging to four local goat breeds. This study supplied the knowledges of the relationship between genomic variations and phenotypic traits.
However, the workload cannot meet the publication requirement. The method used in this manuscript is just an elementary research on CNV analysis. Furthermore, the CNVs identified by chip data always present high false positive for its low resolution. From the perspective of innovation, there was no new approach and significant findings in this study.
RESPONSE: Although many studies of CNVs are available for cosmopolitan breeds, no information was available until now for local breeds such as the Sicilian local goat breeds. The present study is the first in which SNP data was used to detect CNVs in Sicilian breeds. Our results provide significant information for the construction of a more complete CNV map of the goat genome and offer an important resource for the investigation of genomic changes in traits of interest in the local goat breeds. Therefore, our results should be of value for future studies, and constitute a preliminary report on the distribution of CNVs in local goat genomes.
We are aware that SNP genotyping data could present high false positive results for its resolution and we stressed this concept in the Introduction of the revised manuscript. Nevertheless, we used PennCNV software to analyze our data as it is defined by several authors, the most reliable and accurate algorithm in detecting CNVs from Illumina BeadChip data. Lines 50-60.
The results can be further improved by adding the comparison of the breed-differential CNVs to reveal the breed differences and qPCR validation of CNVs. The association analysis between the CNVs and phenotypic traits can also be included in this manuscript if the phenotype data available.
RESPONSE: As requested by other reviewers, we reported in Supplementary Material the information on the frequency of each CNVR within each breed and we showed how many animals per breed carried it (Table S4 and Table1). In the revised manuscript, we highlighted the CNVRs sharend among breeds (Fig 3A) and performed a PCA analysis to revelaed breed differences (Fig. 3B). Lines 122-138, 183-200.
We are aware that novel CNVs should be validated by qPCR, but some scientific references (e.g. Xu et al., 2016 – doi:10.1038/srep23161; de Lemos et al., 2017 – doi:10.1016/j.livsci.2017.11.08; Strillacci et al., 2018 – doi:10.1371/journal.pone.0204669; Di Gerlando et al., 2018 -doi.org/10.1071/AN17603) did not perform validation considering that CNV maps are already published and available from previous studies.
Our findings just confirmed the presence of copy number variations described by other authors in previous studies. Lines 212-258.
As we reported in the Introduction of the revised manuscript, selection schemes are absent for these breeds and milk production or other phenotypic traits are not recorded, therefore, it is not possible to perform association analysis. Lines 73-82

Reviewer 3 Report
The authors performed a very interesting study on local goat populations, providing an overview of genomic biodiversity of Sicilian goat breeds.
The manuscript needs to be improved to be accepted for publication. Details of CNV/CNVR proper of each breed are missing and must be included in the manuscript to make it comparable with other published paper and, as such, give value to the manuscript.
A brief description of each breed in Introduction or Materials and Methods section should be added to give readers necessary information on the populations analyzed for this study.
Results: The following results and analyses must be added before acceptance:
- A table with Descriptive Statistics of CNVs and CNVRs identified for each breed, reporting also the number of samples resulted after filtering;
- A table of CNVRs identified in at least 2 samples, in which are reported the number of samples having that CNVs, something like:
|
CNVR_ID |
START |
END |
STATE |
BREED1 |
BREED2 |
BREED 3 |
BREED4 |
|
|
|
|
|
|
|
|
|
Indicate the CNVR resulted different in different state.
- To make the manuscript comparable with other studies the authors should also add a column with indication of CNVRs identified in other goat breeds. This allow to provide strength to this manuscript and a fundamental rationale for discussion.
- Also, the authors have to include at least a Principal Component Analysis or a clustering one to highlight differences among breeds. This is a fundamental result to present in order to have a general overview of the existing genomic variability based on CNV in the analysed breeds.
-
"Annotation of the most representative CNVRs which had frequency >0.10", maybe io too stringent. The reviewer suggests 0.05%.
- Information on genes and relations to mapped QTL should be included. the authors should consult the QTL genome database (using cattle as species reference – a ruminant species so somehow comparable) using the option “Search by associated gene” (https://www.animalgenome.org/cgi-bin/QTLdb/BT/search), to provide information about gene function.
All the discussions have to rewritten according to the new results that author will include in the revised manuscript.
Author Response
Reviewer 3
Comments and Suggestions for Authors
The authors performed a very interesting study on local goat populations, providing an overview of genomic biodiversity of Sicilian goat breeds.
The manuscript needs to be improved to be accepted for publication. Details of CNV/CNVR proper of each breed are missing and must be included in the manuscript to make it comparable with other published paper and, as such, give value to the manuscript.
RESPONSE: Thanks for the general comments and suggestions. As requested by other reviewers, we reported in the revised manuscript (as Supplementary Material) the information on the frequency of each CNVR within each breed and how many animals per breed carried it (Table S4 – Table 1). Moreover, it is difficult to directly compare CNV results across different methods. For example, the CNV discovery method of PennCNV software tends to focus on finding long and/or rare CNVs while other methods, like the multivariate one, considering all samples simultaneously, were designed for detecting small and common CNVs [Bhanuprakash et al., 2018; Liu et al., 2020]. Also, the difference for breeds and platforms could be other factors for differences in CNV detection results [Liu et al., 2020]. Nevertheless, we reported information about the CNVRs previously identified by other authors (Reference column in Table S6) and discuss it in the manuscript. Lines 212-258.
A brief description of each breed in Introduction or Materials and Methods section should be added to give readers necessary information on the populations analyzed for this study.
RESPONSE: We reported in the Materials and Methods section of the revised manuscript a brief description of each breed. Lines 95-101.
Results: The following results and analyses must be added before acceptance:
- A table with Descriptive Statistics of CNVs and CNVRs identified for each breed, reporting also the number of samples resulted after filtering;
- A table of CNVRs identified in at least 2 samples, in which are reported the number of samples having that CNVs, something like:
|
CNVR_ID |
START |
END |
STATE |
BREED1 |
BREED2 |
BREED 3 |
BREED4 |
RESPONSE: As reported above, we added information on the frequency of CNVRs within breeds and we showed how many animals per breed carried them (Table S4). Moreover, in the revised manuscript we reported Table 1 containing CNVRs descriptive statistics for each breed and overall.
Indicate the CNVR resulted different in different state.
- To make the manuscript comparable with other studies the authors should also add a column with indication of CNVRs identified in other goat breeds. This allow to provide strength to this manuscript and a fundamental rationale for discussion.
RESPONSE: We added a column in Tables S6 with the indication of the papers in which CNVRs were identified in other goat breeds and we reported the indication in the revised manuscript. Lines 212-258. - Also, the authors have to include at least a Principal Component Analysis or a clustering one to highlight differences among breeds. This is a fundamental result to present in order to have a general overview of the existing genomic variability based on CNV in the analysed breeds.
RESPONSE: We thank the reviewer fotr the suggestion. We performed PCA to highligth genetic differences among breeds using a CNVRs matrix containing the indication of CNV in the CNVRs as normal state, duplication or deletion (Table S5). Moreover, we included a Venn diagram to highlight the CNVRs shared among breed and proper of each of them. Lines 131-138, 183-200. - "Annotation of the most representative CNVRs which had frequency >0.10", maybe io too stringent. The reviewer suggests 0.05%.
RESPONSE: We preferred do not change the frequency threshold to make our results more robust expecially because we collected few animals per breed and we applied very stringent filters. - Information on genes and relations to mapped QTL should be included. the authors should consult the QTL genome database (using cattle as species reference – a ruminant species so somehow comparable) using the option “Search by associated gene” (https://www.animalgenome.org/cgi-bin/QTLdb/BT/search), to provide information about gene function.
RESPONSE: Thanks for the suggestion, we consulted QTL genome database as suggested but no associated gene was found. Nevertheless, the function of genes within CNVRs was reported in the revised manuscript.
All the discussions have to rewritten according to the new results that author will include in the revised manuscript.
RESPONSE: We reported in the revised manuscript the requested changes.

Round 2
Reviewer 2 Report
This manuscript has supplemented several results to improve the content,including MDS plot, the information on the frequency and the comparison of breed-differential CNVRs and PCA analysis. It is noteworthy that the result of PCA analysis in Fig.3B is not sufficient to distinguish among the four breeds. I suggest authors adopt other method or optimize PCA to show the breed differences.
The manuscript needs to have a minor revision before publication.
Author Response
Reviewer 2
Comments and Suggestions for Authors
This manuscript has supplemented several results to improve the content, including MDS plot, the information on the frequency and the comparison of breed-differential CNVRs and PCA analysis. It is noteworthy that the result of PCA analysis in Fig.3B is not sufficient to distinguish among the four breeds. I suggest authors adopt other method or optimize PCA to show the breed differences.
The manuscript needs to have a minor revision before publication.
RESPONSE: We thank the reviewer for the general comment and suggestions. We wold highlight that the MDS and PCA plots were obtained using different datasets and for different purposes.
In particular, the MDS plot was obtained using SNP genotyping data of thousands of loci and all samples. This analysis was requested by Reviewer 1 aiming to examine and visualize the genetic relationships among Sicilian goat breeds.
The PCA analysis in Figure 3 was requested by Reviewer 3 in order to compare the distribution of CNVRs among breeds and to show the breed differences related to CNVRs. For this analysis the dataset contained 75 CNVRs and 107 individuals. In this case, the distance among breeds is based on both the major or minor number of shared CNVRs and the state of each CNVR in term of absence, deletion or duplication of CNV events. Line 134

Reviewer 3 Report
The manuscript has been improved according to the suggestions of reviewer.
Please to modify the Figure 3. Increase the size of dots. Past software allows to increase them.
Author Response
Reviewer 3
Comments and Suggestions for Authors
The manuscript has been improved according to the suggestions of reviewer.
Please to modify the Figure 3. Increase the size of dots. Past software allows to increase them.
RESPONSE: We thank the reviewer for the comment. We increased the dots in Figure 3 as requested.
